# Assessment of Temporal Effects of a Mud Volcanic Eruption on the Bacterial Community and Their Predicted Metabolic Functions in the Mud Volcanic Sites of Niaosong, Southern Taiwan

**DOI:** 10.3390/microorganisms9112315

**Published:** 2021-11-08

**Authors:** Ho-Chuan Hsu, Jung-Sheng Chen, Viji Nagarajan, Bashir Hussain, Shih-Wei Huang, Jagat Rathod, Bing-Mu Hsu

**Affiliations:** 1Department of Medical Imaging, Cheng Hsin General Hospital, Taipei City 112, Taiwan; hsuhochuan@gmail.com; 2Department of Medical Research, E-Da Hospital, Kaohsiung City 824, Taiwan; nicky071214@gmail.com; 3Department of Earth and Environmental Sciences, National Chung Cheng University, Chiayi County 621, Taiwan; mathumitha08@gmail.com (V.N.); bashir.aku@gmail.com (B.H.); 4Department of Biomedical Sciences, National Chung Cheng University, Chiayi County 621, Taiwan; 5Center for environmental Toxin and Emerging Contaminant Research, Cheng Shiu University, Kaohsiung City 824, Taiwan; envhero@gcloud.csu.edu.tw; 6Super Micro Research and Technology Center, Cheng Shiu University, Kaohsiung City 824, Taiwan; 7Department of Earth Sciences, National Cheng Kung University, Tainan 701, Taiwan; jagat2006@gmail.com

**Keywords:** mud volcano, sulfur-reducing bacteria, methanogens, hydrocarbon degraders, 16S rRNA gene sequencing, PICRUSt2

## Abstract

The microbial communities inhabiting mud volcanoes have received more attention due to their noteworthy impact on the global methane cycle. However, the impact of temporal effects of volcanic eruptions on the microbial community’s diversity and functions remain poorly characterized. This study aimed to underpin the temporal variations in the bacterial community’s diversity and PICRUSt-predicted functional profile changes of mud volcanic sites located in southern Taiwan using 16S rRNA gene sequencing. The physicochemical analysis showed that the samples were slightly alkaline and had elevated levels of Na^+^, Cl^−^, and SO_4_^2−^. Comparatively, the major and trace element contents were distinctly higher, and tended to be increased in the long-period samples. Alpha diversity metrics revealed that the bacterial diversity and abundance were lesser in the initial period, but increased over time. Instead, day 96 and 418 samples showed reduced bacterial abundance, which may have been due to the dry spell that occurred before each sampling. The initial-period samples were significantly abundant in haloalkaliphilic marine-inhabiting, hydrocarbon-degrading bacterial genera such as *Marinobacter*, *Halomonas*, *Marinobacterium,* and *Oceanimonas*. Sulfur-reducing bacteria such as *Desulfurispirillum* and *Desulfofarcimen* were found dominant in the mid-period samples, whereas the methanogenic archaeon *Methanosarcina* was abundant in the long-period samples. Unfortunately, heavy precipitation encountered during the mid and long periods may have polluted the volcanic site with animal pathogens such as *Desulfofarcimen* and *Erysipelothrix*. The functional prediction results showed that lipid biosynthesis and ubiquinol pathways were significantly abundant in the initial days, and the super pathway of glucose and xylose degradation was rich in the long-period samples. The findings of this study highlighted that the temporal effects of a mud volcanic eruption highly influenced the bacterial diversity, abundance, and functional profiles in our study site.

## 1. Introduction

Mud volcanoes are remarkable geological structures formed by the eruption of high-pressure mud or slurries, water, and gas from the Earth’s deep subsurface [1,2]. Among them, gas is a potential driving force, which pushes mud from deep underground towards the surface through a geological fault or fissure. The breccia, slurries, and gas emitted from the mud volcanoes are geologically connected to deep surface petroleum and natural gas reservoirs, thus providing a reach to probe the characteristics of those deep sources [3]. Mud volcanoes are typically cone-shaped circular structures, from which fluidized mud or slurries are expelled through seepages or explosions. They are geographically localized in areas of recent tectonic activity, especially in zones of compression [1]. Methane and carbon dioxide constitute the major gaseous phase accompanying the emission of fluids from most mud volcanoes.

Currently, studying the microbial communities existing in mud volcanoes has received more attention due to their impact on the global methane cycle [4,5]. The mud volcanic fluids ejected from the reservoir may be geothermally heated; thus, the microorganisms with higher temperature and salinity tolerance can thrive better in these environments [4,6]. Strictly anaerobic methanogens have been reported in mud volcanoes worldwide, which implied that along with the geological phenomenon, the biological process of mud volcanoes might lead to methane emission [4,6,7]. Apart from methane emission, sulfate reduction is one of the key phenomena associated with mud volcanic sites [7]. Sulfate reduction is an anaerobic process that utilizes sulfate as a terminal electron acceptor, which is facilitated by the sulfate-reducing bacteria (SRB) having a key part in both the sulfur and carbon cycles of volcanic sites [8]. Besides methanogens and SRB, a consortium of methanotrophic archaea with aliphatic and polyaromatic hydrocarbon oxidizers, as well as syntrophic iron and nitrate reducers, have been isolated in those anoxic conditions, showing a remarkable diversity of heterotrophic microorganisms in mud volcanic sites [9,10]. In particular, the mud volcanic region was composed of both aerobic organotrophs such as *Halomonas*, *Marinobacterium*, and *Marinobacter,* as well as fermentative microbes such as *Pelobacter* [4,6]. In particular, *Halomonas*, a chemoorganotrophic halophilic bacterium, is typically found in mud volcanoes with higher salinity levels [4]. In addition, hydrocarbon-degrading multiple enzyme producers such as *Marinobacter* have been recovered from the mud volcanic zones of petroleum deposits [11].

Many of the onshore and offshore mud volcanoes have been documented in Taiwan [12,13,14,15]. In particular, mud volcanoes located in the southern part of Taiwan were found to be continuously emitting muddy fluids, slurries, and gases such as methane [14,16,17]. Most of the mud volcanoes in southern Taiwan are categorized as high saline spots, and those mud liquids are reported to have higher Na^+^ and Cl^−^ levels [13,14]. Prior studies conducted in southern Taiwan demonstrated the association between the biogeochemical process of the mud volcanoes with the microbial consortium [6,18]. Another study integrated geochemical and microbial community patterns to address the potential syntrophic relationship between methanotrophs and metal-respiring metabolism in mud volcanoes [19]. The geographic characteristics and the interaction of specific microbial consortiums with the biogeochemical processes of mud volcanoes are well understood [3,18]; nonetheless, the overall microbial diversity, abundance, and the influence of temporal effects of mud fluid eruption on the diversity and potential metabolic functions of the existing microbes are rare and little known.

In this context, this study aimed to uncover and compare the bacterial diversity, abundance, and functional profile changes of the mud volcano located in Niaosong, Kaohsiung, Southern Taiwan. To gain insights into the impact of temporal variation on the microbial diversity, we collected volcanic fluid samples at the initial, mid, and long periods after the volcanic eruption. We analyzed the physicochemical characteristics of the collected mud volcanic samples at different time intervals. Furthermore, the bacterial community structure and their functional predictions were determined based on 16S rRNA amplicon sequencing. This study provided a substantial characterization of the mud volcano located in Niaosong, as well as their bacterial abundance and functional profiles. In addition, this study demonstrated the influence of temporal variability of mud volcanic eruption on bacterial abundance and their potential metabolic functions.

## 2. Materials and Methods

### 2.1. Sampling Site Description and Sample Collection

Our sampling site was located in Niaosong District, Kaohsiung City, Taiwan (22°39′37.7″ N and 120°22′01.3″ E). The geographical locations of the sampling sites are shown in Figure 1. The mud fluid samples were collected right below the gas bubbling area using four 50 cm^3^ preweighted polypropylene (PP) centrifuge tubes. Mud samples were collected on 12 February 2016 at different successional stages after the volcanic eruption, such as initial (day 1, 4, and 8), mid (day 22 and 96), and long period (day 158 and 418), for physicochemical and molecular analysis. On days 96 and 418 of sample collection, the sample was too dry due to zero precipitation for 20 days before sampling. On the contrary, at days 22 and 158 of sample collection, the site experienced two rainy days just before each sampling. The temperature, pH, and oxidation/reduction potential (ORP) values were obtained onsite with a WalkLAB^®^ TI9000 temperature compensation pH meter (Trans Instruments, Singapore). The retrieved samples were labeled accordingly and transported to the laboratory under controlled temperature for subsequent analysis.

### 2.2. Processing and Physicochemical Analysis

In the laboratory, the samples were filtered through 0.45 μm nylon membrane filters then centrifuged at 2560× *g* for 30 min. The residual solids were oven-dried at 50 ℃ overnight. The weight of dried mud with a centrifuge tube was measured. The percentage of mud weight was then obtained by the dry weight over raw weight after deducting the weight of the centrifuge tube. The filtered samples were separated into two aliquots, with one being acidified with concentrated nitric acid to pH < 2 and used for trace elemental analysis. The other remaining nonacidified samples were preserved for the measurement of cations, anions, and total alkalinity. Then the samples were kept at 4 ℃ in the refrigerator for further analysis.

Major and trace element concentrations (S, Na, Ca, K, Mg, B, Sr, Ba, and Si) were measured with inductively coupled plasma optical emission spectrometry (ICP-OES) with a precision of better than 3% (Agilent 510, Santa Clara, CA, USA). Dissolved anions (Cl^−^) were determined with ion chromatography (Dionex^®^ ICS-3000, Thermo Scientific, Waltham, MA, USA) with a precision better than 5%. Accuracy was checked in each batch using the diluted international seawater standard of the International Association for the Physical Sciences of the Oceans (IAPSO). The chloride concentration of the diluted IAPSO standard was calculated based on salinity and the equation formulated by a past study [1]. Total alkalinity (TA) was measured by acid titration (Metrohm^®^ 905 Titrando, Metrohm AG, Herisau, Switzerland) with a precision better than 1%, estimated by repeated analyses of the sample (n = 3) and an in-house-prepared bicarbonate standard.

### 2.3. DNA Extraction and 16S rRNA Amplicon Sequencing

The processed soil samples were mixed thoroughly, and 0.5 g of fine soil samples were used for genomic DNA extraction using a commercial soil gDNA extraction kit, the NucleoSpin^®^ kit for soil (Macherey-Nagel GmbH & Co., Düren, Germany), following the manufacturer’s protocol. Subsequently, the concentrations and the purity of the extracted gDNA sample were checked using a Nanodrop 1000 spectrophotometer (Thermo Fisher Scientific, Waltham, MA, USA) at 260/280 nm. The purified gDNA was then stored at −20 ℃ for further analysis.

From the gDNA extracts, the hypervariable V3-V4 region of the16S rRNA gene was amplified using the following primer sets: Pro341F (5′-CCTACGGGNBGCASCAG-3′) and Pro805R (5′-GACTACNVGGGTATCTAATCC-3′). PCR was carried out in 25 μL reactions including 3 μL of gDNA, 12.5 μL of 2X KAPA HiFi HotStart Ready Mix, 10 M of each primer, and 7.5 μL of ddH_2_O. The PCR reaction was carried out in a thermocycler (Px2 Thermal Cycler, Thermo, Waltham, MA, USA) under the following conditions: denaturation at 95 °C for 30 s, 28 amplification cycles of 30 s at 95 °C, annealing at 55 °C for 30 s, and a final extension at 72 °C for 30 s. PCR products were visualized using gel electrophoresis, and the expected sizes of PCR products were purified from the matrix. The amplicons were sequenced using the pair-end method with the MiSeq Illumina platform (Illumina Inc., San Diego, CA, USA), following the manufacturer’s instructions.

### 2.4. Library Construction, Data Analysis, and Metabolic Functional Prediction

The Nextera XT DNA sample preparation kit (Illumina) was used to construct DNA libraries. The raw sequence files from the NGS platform were then processed using the Quantitative Insights into Microbial Ecology (QIIME2) pipeline. After removing the chimeric, noisy sequences, and marginal sequence errors, amplicon sequence variants (ASVs) were picked using DADA2, and further taxonomy classification was done using the SILVA reference database [20]. The relative abundance of the microbial community associated with each sample was obtained using the QIIME2 view [21]. The potential metabolic functions of the bacterial community at different time intervals after the volcanic eruption were determined using the latest version of the PICRUSt2 pipeline using the MetaCyc Metabolic Pathway Database (https://metacyc.org/, accessed on 10 August 2021).

### 2.5. Correlation Analysis

The significant shift in the bacterial community among the samples at various time intervals at the phylum and genus levels was analyzed using a two-tai ㄝ led Welch’s *t*-test with STAMP software (*p* < 0.05). A Spearman correlation analysis was carried out to determine the significant correlation between the bacterial community and the metagenomic functional prediction.

## 3. Results

### 3.1. Physicochemical Characteristics of the Mud Samples

The physicochemical characteristics of the mud samples at different time intervals after the eruption are shown in Table 1. The pH values denoted that the muddy fluid collected at the initial, mid, and long periods were slightly alkaline (7.53–8.42). The samples on days 1 and 158 after the volcanic eruption showed a negative oxygen reduction potential (ORP); however, the remaining samples recorded a positive ORP. The major cation was sodium, followed by calcium, potassium, boron, and magnesium; and the major anion was chlorine, followed by sulfate. The total alkalinity of the samples had an increasing trend with the days following volcanic eruption, but the value suddenly declined in the long-period samples. The concentrations of Na^+^ and Cl^−^ ions in the samples were clearly elevated (Na^+^: 7538–13,301 ppm; Cl^−^: 10,791–18,343 ppm) compared to other ions such as SO_4_^2−^, Ca^2+^, K^+^, Mg^2+^, and other minor elements, suggesting that the origin of salinity may have been from a marine source. Comparatively, the long-period samples recorded higher Na^+^ and Cl^−^ content than those of the initial and mid periods. Similarly, SO_4_^2−^ ions and total sulfur content were distinctly higher, and were observed to be increased in the later days following the volcanic eruption. The Ca^2+^/Cl^−^, Mg^2+^/Cl^−^, and SO_4_^2−^/Cl^−^ ratios of the samples were in the ranges of 0.002–0.010, 0.002–0.003, and 0.0007–0.0048, respectively. The concentrations of other elements such as B, Sr, Ba, and Si were lower than those of the major elements.

### 3.2. Diversity Metrices of the Bacterial Communities

Alpha diversity was used to analyze the bacterial community’s diversity and richness within the individual samples collected at different time intervals after the volcanic eruption (Figure 2). The diversity metrics of the initial-, mid-, and long-period samples were estimated using Observed_OTUs and the Chao1 index. The results demonstrated that higher diversity richness was associated with the long-period samples and a decreased diversity was observed in the mid-period samples, while the lowest diversity metrics were noted in the initial-period samples. However, none of these indices showed significant diversity richness among the samples collected at different intervals. Furthermore, beta diversity based on a principal coordinate analysis (PCoA) enabled a direct comparison of bacterial communities between the samples. The results evidenced a higher variation among the samples collected at different time intervals, indicating different bacterial communities associated with the three sampling periods. The PC1 of ordination representing the initial period showed the highest variation (PC1: 43.3%), followed by the long period (PC2: 34.2%), as compared to the mid period (PC3: 11.6%). Among them, a distinct clustering pattern was noticed in the samples collected during the initial period. 

### 3.3. Diversity and Relative Abundance of the Bacterial Communities

#### 3.3.1. Phylum Level Diversity and Relative Abundance of the Bacterial Communities

A total of 14 classifiable bacterial phylum were generated based on the 16S rRNA amplicon sequencing taxonomic classification (Figure 3A). The number of abundant phyla noted in the mid and long periods were comparatively higher than the initial-period samples. Particularly, the samples collected at 22 and 158 days after the volcanic eruption exhibited a higher number of abundant phylum than the other samples. On the contrary, only a few phyla were noticed in the samples at 96 and 418 days after the eruption. Proteobacteria were predominant in all the samples, irrespective of their collection period. Firmicutes was the second most abundant phylum, and it showed higher dominance in the mid- and long-period samples than in the initial-period samples. In particular, Desulfobacterota was abundant only in the initial-period samples, especially in the samples collected on the next day after the volcanic eruption. Chrysiogenetota and Bacteroidota exhibited their abundance only in the mid- and long-period samples. However, Spirochaetota was found only in the long-period samples collected on day 158.

The changes in bacterial diversity and abundance at the phylum level in the samples were examined using STAMP software (Figure 3B). The analysis revealed that the abundance of the phylum Proteobacteria was significantly enriched between the initial and mid periods, with a higher abundance in the initial-period samples, while the mean proportion of the phylum markedly decreased in the mid-period samples. Likewise, the phylum Firmicutes was significantly enriched between the initial- and long-period samples. The phylum was predominant in the long period, whereas the mean proportion of this phylum decreased in the initial period.

#### 3.3.2. Genus-Level Diversity and Relative Abundance of the Bacterial Communities

A total of 96 classifiable bacterial genera were generated based on the 16S rRNA amplicon sequencing taxonomic classification (Figure 4). Principal component analysis (PCA) was done to analyze the variation in the bacterial community compositions at the genus level among the samples collected at various time intervals after the volcanic eruption (Figure 4A). PCA analysis exhibited the obvious distribution pattern and uniqueness of the samples, and also that all the samples were distinctly separated from each other. PC1, PC2, and PC3 accounted for 41.1%, 32.1%, and 11.4% of the total variation, respectively. This indicated the uniqueness of bacterial communities at the genus level. The initial-period samples showed a distinct variation in the bacterial community’s composition, indicating the existence of distinct bacterial genera in the initial-period samples. However, the mid- and long-period samples had fewer variations, indicating less dissimilarity in their bacterial diversity at the genus level.

In addition, genus-level comparative analyses showed that the bacterial genus abundances were expressively different between the samples collected at various time intervals (Figure 4B). Comparatively, a higher bacterial diversity and abundance was noted in the samples collected on days 22 and 158 after the volcanic eruption. On the contrary, less bacterial diversity was noticed in the samples collected on days 1, 96, and 418. The genera *Marinobacter* and *Halomonas* were found abundant in the samples collected at the initial and mid periods, but not in the long-period samples. Considering the initial-period samples, *Marinobacterium* and *Idiomarina* were found rich in the samples collected on days 4 and 8. Regarding the mid-period samples, the genera *Marinospirillum* and *Comamonas* were found abundant on day 96. In the long-period samples, *Sphingomonas* was found abundant in the sample from day 418. In addition, the genus *Anaerobacillus* was found predominant in the samples collected on days 96 and 418. Some of the genera, such as *Desulfurispirillum*, *Desulfofarcimen*, *Acholeplasma*, and *Marivirga,* were found only in the samples from days 22 and 158, but not from other days. Specifically, the genera *Bowmanella* and *Pelospora* were found dominant only on day 22; and *Erysipelothrix*, *Bradymonadaceae*, *Rhodohalobacter*, *Methanosarcina*, *Serpentinicella*, and *Sediminispirochaeta* were predominant in the samples from day 158. 

Additionally, the changes in bacterial diversity and abundance at the genus level in the samples from different intervals were examined using STAMP software (Figure 5). The comparative analysis revealed that the abundance of *Halomonas* and *Marinobacter* were significantly different between the initial- and mid-period samples, and the abundance was enriched in the initial-period samples. The mean proportion of the genera significantly decreased in the mid period. Likewise, *Halomonas*, *Marinobacter*, *Pseudomonas*, *Marinobacterium*, *Oceanimonas*, and *Pelospora* were significantly different between the initial- and long-period samples, and their abundance was enriched in the initial-period samples. However, the mean proportion of the above-mentioned genera markedly decreased in the long-period samples.

### 3.4. Predictive Functional Analysis of the Bacterial Community

A total of 360 pathways were detected based on 16S rRNA by using the MetaCyc database among the three experimental groups. Principal component analysis (PCA) was carried out to analyze the variation in the predictive functions of bacterial communities collected at various time intervals (Figure 6). PC1, PC2, and PC3 accounted for 42.1%, 27.9%, and 15.8% of the total variation, respectively. Apparently, the samples from each of the experimental groups were separated from each other, which indicated their uniqueness in the potential metabolic functions. Particularly, the initial-period samples showed a distinct variation, indicating the existence of unique predicted functions in those samples.

#### 3.4.1. Shift Analysis of the Predicted Functions of the Bacterial Community

The STAMP software was used to further investigate significant changes in the metabolic functions of the samples collected at different time intervals after the volcanic eruption (Figure 7). The comparative analysis revealed that the predicted metabolic function ((5Z)-dodec-5-enoate biosynthesis) was significantly enriched (*p* < 0.05) between the initial- and mid-period samples (Figure 7A). The mean proportion of the aforementioned function was decreased in the mid-period samples. Additionally, 10 predicted metabolic functions, such as (5Z)-dodec-5-enoate biosynthesis, palmitoleate biosynthesis I, oleate biosynthesis IV, 4-deoxy-L-threo-hex-4-enopyranuronate degradation, super pathway of glucose and xylose degradation, stearate biosynthesis II, ubiquinol-7 biosynthesis, ubiquinol-8 biosynthesis, ubiquinol-10 biosynthesis, and ubiquinol-9 biosynthesis, were significantly enriched between the initial- and long-period samples (Figure 7B). Particularly, the abundance of (5Z)-dodec-5-enoate biosynthesis, palmitoleate biosynthesis I, oleate biosynthesis IV, stearate biosynthesis II, ubiquinol-7 biosynthesis, ubiquinol-8 biosynthesis, ubiquinol-10 biosynthesis, and ubiquinol-9 biosynthesis in the initial-period samples was higher than that in the long-period samples. However, the abundance of two functions, 4-deoxy-L-threo-hex-4-enopyranuronate degradation and super pathway of glucose and xylose degradation, was higher in the long period than in the initial samples.

#### 3.4.2. Correlation between the Bacterial Genus and Predicted Metabolic Functions 

Spearman correlation analysis was carried out to obtain the relationship between statistically enriched bacterial genera and their predicted metabolic functions (Figure 8). The association were considered significant at *p* < 0.05. The correlation analysis revealed that *Marinobacter* and *Halomonas* were significantly and positively correlated (*p* < 0.05) with (5Z)-dodec-5-enoate biosynthesis, palmitoleate biosynthesis I, oleate biosynthesis IV, stearate biosynthesis II, ubiquinol-7 biosynthesis, ubiquinol-8 biosynthesis, ubiquinol-10 biosynthesis, and ubiquinol-9 biosynthesis. None of the identified genera, except *Marinobacter* and *Halomonas*, were significantly and positively correlated with the predicted functions related to the mud volcanic sites. However, *Marinobacterium* showed a positive correlation with all the predicted functions, but it was not significant.

#### 3.4.3. Ordination Analysis

The principle coordinate analysis (PCoA) triplots showed a visual interpretation of the relationship between the bacterial genera, functional pathways, and physicochemical properties with the periods of sample collection (Figure 9). The genera *Oceanimonas* and *Pseudomonas* (day 4), and *Marinobacterium*, *Halomonas*, *Marinobacter* (day 8), were abundant in the initial-period samples, whereas *Pelospora* abounded in the mid-period samples (days 22 and 96). The genera such as *Marinobacter*, *Halomonas*, and *Marinobacter* were highly positively associated with functional pathways such as dodec-5-enoate biosynthesis, palmitoleate biosynthesis I, oleate biosynthesis IV, stearate biosynthesis II, ubiquinol-7 biosynthesis, ubiquinol-8 biosynthesis, ubiquinol-10 biosynthesis, and ubiquinol-9 biosynthesis. The aforementioned functional pathways were highly correlated with the initial-period samples. In contrast, the genus *Pelospora* was negatively correlated with the above-mentioned predicted functions, but it was positively correlated with the physicochemical properties such as pH, TS, TDS, Na, Mg, K, B, and Ba content. In addition, the samples collected on days 1, 22, and 96 were negatively correlated with all the predicted functions.

## 4. Discussion

The geographical location and biogeochemical processes that occur in volcanic sites can influence the physicochemical characteristics of muddy fluids of mud volcanic areas [22]. The mud volcanic samples collected at the Niaosong site had similar physicochemical characteristics to those of seawater, such as slight alkalinity and elevated levels of ions such as Na^+^ and Cl^−^. These findings suggested that the origin of the alkalinity and higher Na^+^ and Cl^−^ levels in mud volcano sites could arise from a marine source [13]. Though Ca^2+^ and Mg^2+^ concentrations were still significant in the muddy samples, the Ca^2+^/Cl^−^ and Mg^2+^/Cl^−^ ratios were lower than those of seawater, which may be attributed to the enhanced activities of sulfate-reducing bacteria that existed at our study site, such as *Desulfurispirillum* and *Desulfofarcimen* [18]. A higher concentration of sulfates was noted in the long-period samples than in the early samples. However, the concentration was much lower than in a marine environment (13.9–283.9 ppm), but it was high enough for the growth of sulfur-reducing bacteria such as *Desulfurispirillum* and *Desulfofarcimen* found at our study site [23]. The higher activity of SRB, which are the primary producers of hydrogen sulfide, in the mid- and long-period samples was measured as total sulfur content in the ICP-OES. Thus, the mid- and long-period samples showed a higher content of total sulfur than the initial-period samples. In addition, a higher content of strontium may indicate an intense dissolution and reprecipitation of carbonates in the fluids [23,24]. 

Alpha diversity metrics such as Observed_OTUs and the Chao1 index demonstrated higher diversity richness associated with the long-period samples, followed by the mid- and initial-period samples. The fluid and gas emissions associated with the mud eruption had a significant impact on environmental variables [25], thereby reducing bacterial diversity and abundance, as noticed in the samples collected immediately after the volcanic eruption (days 1, 4, and 8). The beta diversity results evidenced a higher variation among the samples; in particular, a distinct clustering pattern was noticed in the samples collected during the initial period. A dry spell and heavy precipitation were observed in the mid and long periods of sample collection. Hence, the environmental variation may have influenced the bacterial community composition in the mid- and long-period samples during the initial days.

The phylum diversity and abundance noted in the mid and long periods were comparatively higher than in the samples of the initial period. The day 22 and 158 samples especially exhibited a higher phylum diversity. As already mentioned, those two days experienced heavy precipitation before sampling. Generally, bacteria can be easily dispersed among distant habitats, thereby potentially influencing local bacterial community composition in recipient environments [26]. Moreover, precipitation facilitates the deposition of microbes into new habitats, and thus in and around existing airborne/waterborne bacteria that potentially reach the mud volcanic sites during rain events. Contrarily, a few phyla were noticed in the day 96 and 418 samples. As mentioned earlier, the samples collected at the day 96 and 418 were too dry due to zero precipitation for 20 days before each sampling. During water deficits, the bacterial communities may experience stochastic stress, thus decreasing the bacterial diversity, as well as activity [27]. Among the phyla, Proteobacteria were predominant in all the samples, irrespective of their collection period. This was consistent with previous studies, which showed an abundance of the phylum Proteobacteria in the mud volcanic sites [7,28]. Sequences affiliated with the phylum Firmicutes were significantly enriched in the long-period samples, which encompassed metabolically versatile bacteria, exclusively SRB [29]. This denoted the abundance of SRB in our volcanic study site, especially in the later period after the volcanic eruption. However, Desulfobacterota was abundant in the initial-period samples, especially in the samples collected on day after the volcanic eruption. Members of the genus Desulfobacterota often form consortia with the methanotrophs that inhabit the mud volcanoes [23]. Chrysiogenetota, found in abundance in the mid- and long-period samples, was previously isolated from the hydrothermally active submarine volcanoes [30]. Additionally, Bacteroidota was dominant in the mid- and long-period samples; most of the thermotolerant, halotolerant, and hydrocarbon-decomposing bacterial genera belonging to the phylum Bacteroidetes were isolated in previous studies [31].

Genus-level comparative analyses revealed that higher bacterial diversity and abundance were noted in samples from days 22 and 158, whereas lower bacterial diversity was noticed in the samples from days 1, 96, and 418. The genera *Halomonas*, *Marinobacter*, *Marinospirillum,* and *Pelospora* were abundant in the initial- and mid-period samples, respectively. These genera represent the heterotrophic bacterial community inhabiting the volcanic sites, and are chemoorganotrophic halophilic bacteria dominant in saline environments [11,32,33]. Organic matter produced by the sulfur-oxidizing bacteria that existed in our mud volcanic sites were good enough to support both the aerobic organotrophs such as *Halomonas*, *Marinobacter*, and *Marinospirillum,* and fermentative bacteria such as *Pelospora*. [4]. *Marinobacter,* found to be dominant in the initial days, is a hydrocarbonoclastic bacteria capable of degrading aliphatic hydrocarbons of the volcanic sites. A previous study isolated *Marinobacter* from a variety of niches such as terrestrial mud volcanoes and seawater, and proved that the genus has an association with methylotrophic methanogens [7,34]. *Pseudomonas,* found to be enriched in the initial day’s sample, was one of the inhabitants of mud volcanic sites at various geographical locations [35,36]. This genus was often involved in aliphatic hydrocarbon degradation of the mud volcanic sites having a geographical connection with natural gas reservoirs [37]. Moreover, unlike other sensitive genera, the growth and activity of *Pseudomonas* were not affected by the hydrogen sulfide produced during the activity of SRB, which also resulted in their higher abundance [38]. *Oceanimonas* is a marine Proteobacteria found abundant in the initial day’s samples, and is a key player in polycyclic aromatic hydrocarbon (PAH) degradation of contaminated marine environments. This was previously isolated from the sea, as well as oil-contaminated coastal beach sediments, and is being employed for PAH degradation [39,40]. One of the marine bacteria, *Idiomarina,* was found to be abundant in the initial day’s samples, and was isolated from surface as well as deep seawater samples in previous studies [41,42].

The genus *Comamonas* was enriched in the mid-period samples, and is a natural food source for nematodes inhabiting mud volcanic sites. In a previous study, nematodes were isolated from the mud volcanoes of the Norwegian Sea [43]. The nematodes could colonize and live in symbiosis with chemoautotrophic bacteria of mud volcanic sites, and they could create an extensive nematode field there [44]. The long-period samples were abundant in *Acholeplasma*, which can colonize the guts and hemolymph of insects, as well as mud crab [45,46]. Thus, the results evidenced that the ambience of the study site facilitated the existence of muddy insects and other invertebrates at the site. *Anaerobacillus,* found dominant in the long-period samples, is an anaerobic diazotrophic, alkaliphilic, and moderately halophilic bacterial genus that was isolated from soda lakhs in previous studies [47]. Thus, this bacterial genus can survive better in the elevated alkaline condition of our study site. *Sphingomonas*, widely distributed in marine water, was found abundant in the long-period volcanic samples. A finding of a previous study highlighted the presence of *Sphingomonas* in a hydrocarbon-contaminated environment [48]. Henceforth, the finding proved that our study site may be connected to deep surface petroleum reservoirs, and where *Sphingomonas* may likely serve as a hydrocarbon degrader [48]. *Methanosarcina*, a methanogenic archaeon, was found rich in the long-period samples, and was responsible for methane emissions at our study site. It also is the genus involved in reverse methanogenesis during net methane production. A previous study stated that the genus is often involved in trace methane oxidation (TMO) during net methane production from growth substrates [49,50]. In this process, methane is oxidized into H_2_ and CO_2_, and H_2_ is transferred to sulfate-reducing bacteria, which makes the overall reaction thermodynamically favorable [50,51]. Fortunately, bacteria involved in TMO were closely related to anaerobic methanotrophs (ANME), and they likely share metabolic similarities with AOM metabolism as well [49]. AOM was usually coupled with the presence and activity of dissimilatory SRB [52]. A strict anaerobic and metabolically versatile genus, *Desulfurispirillum,* was abundant in the samples from days 22 and 158, and was responsible for the sulfur- and nitrate-reduction process that occurred in the mud volcanic sites [53,54]. Another sulfur reducer, *Desulfofarcimen,* was found abundantly in the day 158 samples, and is a strict anaerobic moderately thermophilic bacterial genus. A higher sulfate content was recorded in the mid- and long-period samples, which could have enhanced the activity of SRB in those samples. In a previous study, the genus *Desulfofarcimen* was isolated from piggery waste, which evidenced that the study sites had been contaminated with nearby piggery units [55]. The contamination may have been due to the flooding caused by heavy precipitation that occurred just before sampling. *Erysipelothrix,* which was abundant in the long-period samples, causes clinical infections in pigs, poultry, and other animals [56]. As previously mentioned, the flooding caused by the heavy rainfall just before sampling could have led to the deposition of these types of pathogenic bacteria genera in the study site during the long period. In addition, the genus *Marivirga,* found abundant in the long-period samples, is one of the marine bacteria commonly living in wet habitats. A previous study isolated some species of *Marivirga* from the wet estuary mud [57]. The heavy precipitation could develop a wet atmosphere, which in turn helped this particular genus in its better survival. Other anaerobic halophilic genera such as *Bradymonadaceae* and *Rhodohalobacter,* found dominant in the long-period samples, were isolated from marine sediments and saltern, respectively, in previous studies [58,59]. Additionally, *Serpentinicella,* found in the long-period samples, is a novel anaerobic bacterium that was previously isolated from serpentinite-hosted hydrothermal fields [60]. The elevated levels of trace metals, especially strontium noted in the mud volcanic site, may be correlated with the isolation of such bacterial genera from the site [61]. The genus *Spirochaeta,* noted in the long-period samples, are anaerobes isolated from a variety of aquatic habitats, especially the sediments of oceans, which can reduce thiosulfate to sulfide [62].

Generally, methanogenesis facilitates the mineralization of complex molecules such as carbohydrates, lipids, and proteins generated by the primary producers that inhabit the anaerobic niches. The functional prediction results revealed that lipid biosynthesis, such as palmitoleate, stearate, and oleate biosynthesis, was found predominant in the initial days’ samples. The abundance of SRB in our volcanic sites, such as *Desulfurispirillum* and *Desulfofarcimen,* facilitated the lipid biosynthesis process [63]. In addition, another study stated that the presence of lipid biomarkers in an environment was indirectly correlated with the presence of *Desulfurispirillum*, *Desulfofarcimen,* and other SRBs [63]. In particular, oleate and stearate biosynthesis was found abundant in the initial days. Oleic acid and stearic acid concentrations beyond the acceptable limits cause slight inhibition of the activity of methanogens [64], thus the abundance of methanogens such as *Methanosarcina* were noted to be lower in the initial days than in the mid and long periods. Ubiquinol-7, 8, 9, and 10 biosynthesis were enriched in our study sites, especially in the initial-period samples. Ubiquinone plays a vital role in the energy-generation process. It is used to scavenge lipid peroxyl radicals and thereby prevents oxidative stress damage, and is responsible for oxidative stress adaptation of microbial communities present in study sites [65]. While glucose and xylose degradation pathways were enriched in the long-period samples, a previous study indicated that xylose accumulation was higher under sulfidogenic and methanogenic conditions [66]. Further, the accumulated glucose and xylose are degraded into methane and carbon dioxide under the activity of glucose- and xylose-degrading methanogens such as *Methanosarcina* [67,68]. In our study site, *Methanosarcina* was shown in abundance in the long-period samples, thus the super pathway of glucose and xylose degradation was noted to be higher in the long-period samples.

## 5. Conclusions

This study highlighted the physicochemical characterization and temporal effects of a volcanic eruption on the bacterial diversity, abundance, and potential metabolic functions of the mud volcanoes in Niaosong District, Southern Taiwan. The results showed that the samples from our study site had similar chemical characteristics to those of seawater, indicating that they could arise from a marine source. The bacterial diversity and abundance were lesser in the initial period, but increased over time. The initial-period samples were abundant in haloalkaliphilic marine-inhabiting, hydrocarbon-degrading bacterial genera such as *Marinobacter*, *Halomonas,* and *Marinobacterium*. The sulfur-reducing bacterial genera such as *Desulfurispirillum* and *Desulfofarcimen* were found in abundance in the mid-period samples, whereas the methanogenic archaeon *Methanosarcina* was shown as abundant in the long-period samples. Nevertheless, the heavy precipitation encountered during the mid and long periods contaminated the study site with *Desulfofarcimen* and *Erysipelothrix*, which previously were isolated from piggery waste. Lipid biosynthesis and ubiquinol pathways were abundant in the initial days, and the super pathway of glucose and xylose degradation was rich in the long-period samples. Our results clearly showed that the temporal effects of the mud volcanic eruption allowed harboring of various specialized bacterial communities of unique functional pathways in our study site.

## Figures and Tables

**Figure 1 microorganisms-09-02315-f001:**
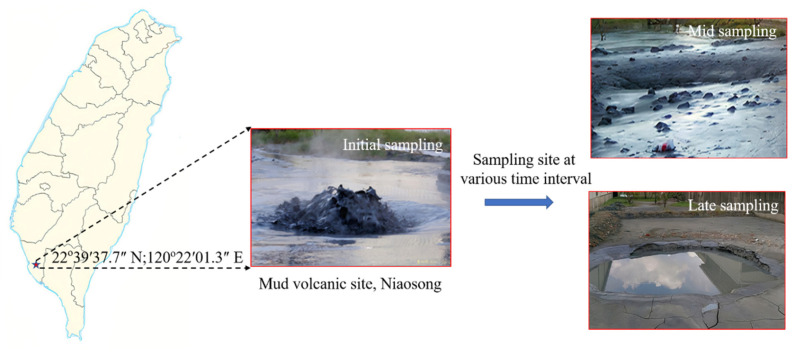
Geographical location and field image of the mud volcanic site, Niaosong District, Kaohsiung, at various sampling intervals.

**Figure 2 microorganisms-09-02315-f002:**
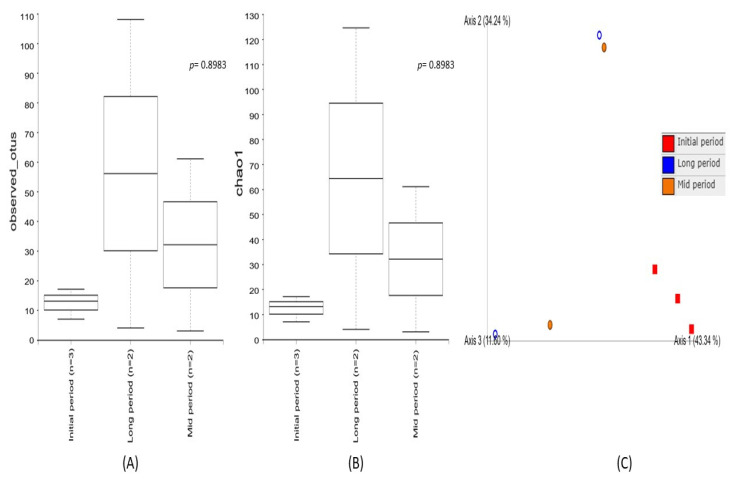
Comparison of bacterial community diversity among the samples collected at the initial, mid, and long periods after volcanic eruption. Box plot of the alpha diversity indices (**A**) Observed_OTUs and (**B**) Chao1 based on 16S rRNA gene amplicon sequencing, demonstrating a higher diversity richness in long-period samples, followed by the mid- and initial-period samples. The line inside the box represents the median, while the whiskers represent the lowest and highest values within the 1.5 interquartile range (IQR). (**C**) Principal coordinate analysis (PCoA) of beta diversity among the samples collected at the initial, mid, and long periods after the volcanic eruption, evidencing a higher variation among the samples.

**Figure 3 microorganisms-09-02315-f003:**
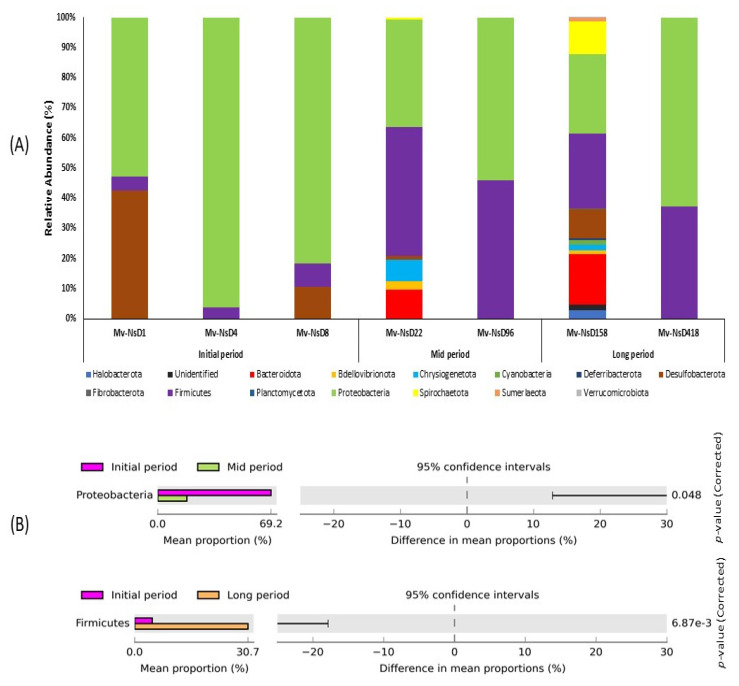
Relative abundance of bacterial diversity at (**A**) the phylum level, showing a comparison between the samples collected at the initial, mid, and long periods after the volcanic eruption. Stacked bar graphs illustrate the abundance of phyla associated with each of the samples. The numbers of abundant phyla in the mid and long periods were higher than in the initial-period samples. (**B**) Post hoc plot of the statistically significant enriched bacterial phyla of the samples collected at various time intervals. The figures show the abundance of the enriched bacterial phylum in the left-side panel, the mean proportion (%) of the enriched bacterial phylum in middle, and the significant difference (*p* < 0.05) in the right-side panel.

**Figure 4 microorganisms-09-02315-f004:**
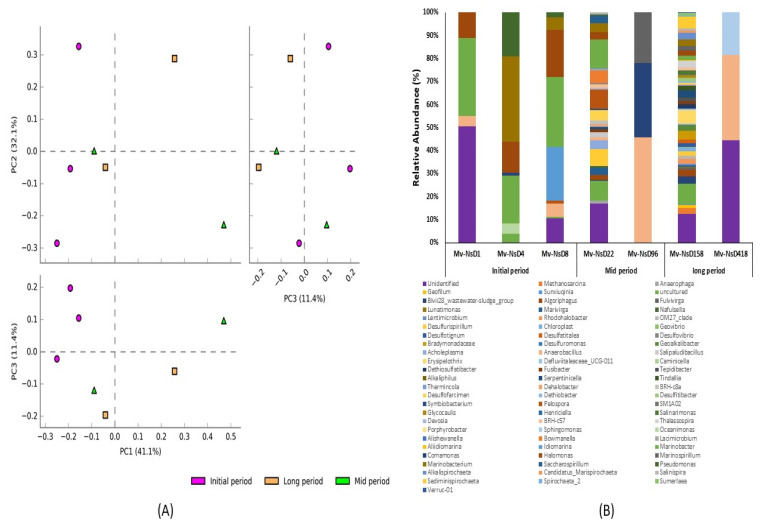
Diversity and relative abundance of bacterial genera of the samples collected at the initial, mid, and long periods after volcanic eruption. (**A**) PCA plots showing the grouping of samples based on the bacterial community structure of the samples collected at various time intervals, which exhibited the distribution pattern and uniqueness of the samples. (**B**) Stacked bar graphs illustrating the abundance of genera associated with the samples collected at various time intervals after the volcanic eruption. Higher bacterial diversity was noted in the samples from days 22 and 158, while lower diversity was observed in the samples from days 1, 96, and 418.

**Figure 5 microorganisms-09-02315-f005:**
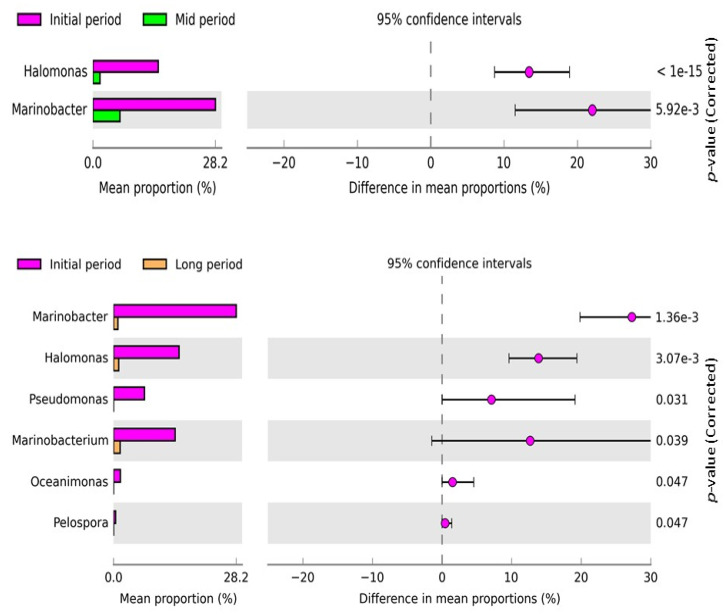
Post hoc plot of the significantly enriched bacterial genera of the samples collected at various time intervals. The figures show the abundance of the enriched bacterial genus in the left-side panel, the mean proportion (%) of the enriched bacterial genus in the middle, and the significant difference (*p* < 0.05) in the right-side panel. Two and six bacterial genera were significantly different between the initial- and mid-period samples and initial- and long-period samples, respectively.

**Figure 6 microorganisms-09-02315-f006:**
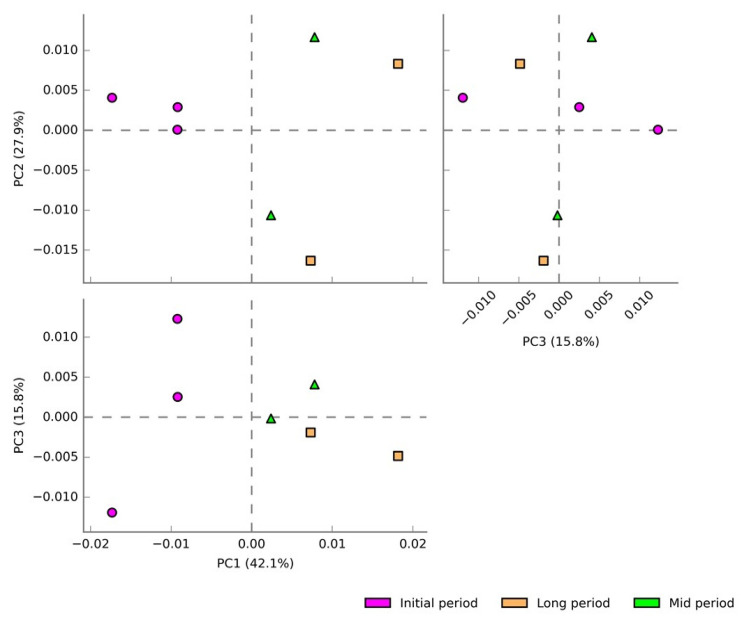
PCA plots showing the grouping of samples based on their predicted metabolic functions collected at various time intervals. The distinct variation indicated their uniqueness in the predicted potential metabolic functions.

**Figure 7 microorganisms-09-02315-f007:**
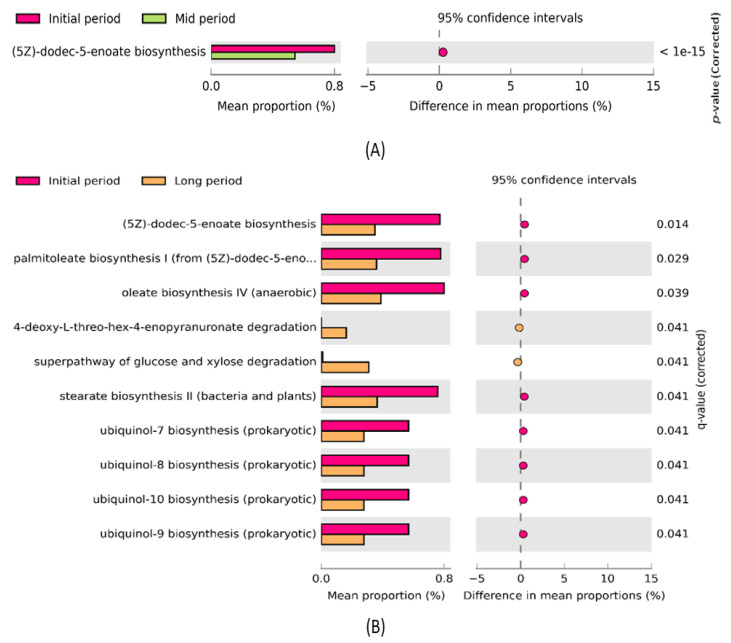
Post hoc plot of significantly enriched predicted metabolic functions among (**A**) initial- and mid-period samples showing significant enrichment of one metabolic function; (**B**) initial- and long-period samples showing significant enrichment of 10 metabolic functions. The figures show the abundance of the enriched predicted metabolic functions in the left-side panel, the mean proportion (%) of the enriched predicted metabolic functions in the middle, and the significant difference (*p* < 0.05) in the right-side panel.

**Figure 8 microorganisms-09-02315-f008:**
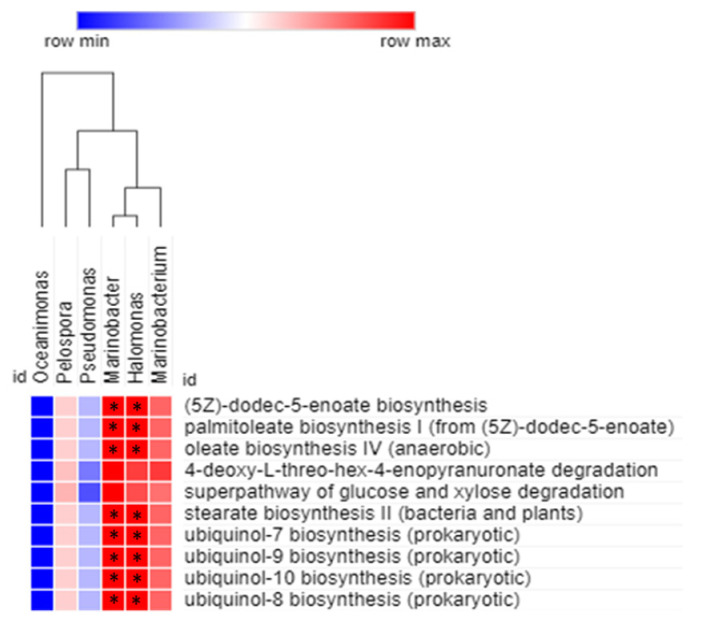
Spearman correlation analysis between bacterial genera of the mud volcanic site and their metabolic functional predictions. The positive and negative correlations are shown in red and blue colors, respectively. The correlation was considered significant at * *p* < 0.05.

**Figure 9 microorganisms-09-02315-f009:**
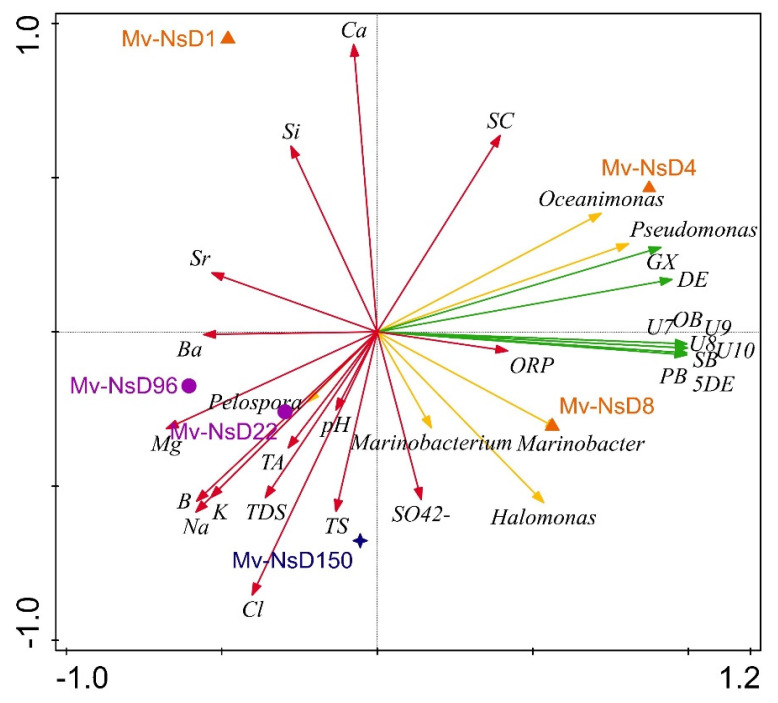
PCoA ordination triplots denoting the relationship of bacterial genera (yellow arrow), functional pathways (green arrow), and physicochemical properties of the samples (red arrow) with the sampling period (orange triangles denote the initial period, purple circles denote the mid period, and blue stars denote the long period) using CANOCO 5. Arrows indicate directions of maximum variation of environmental variables; the length of the arrows indicates their importance. Abbreviations used in the figure are as follows: 5DE: (5Z)-dodec-5-enoate biosynthesis; PB: palmitoleate biosynthesis I; OB: oleate biosynthesis IV; DE: 4-deoxy-L-threo-hex-4-enopyranuronate degradation; GX: super pathway of glucose and xylose degradation; SB: stearate biosynthesis II; U7: ubiquinol-7 biosynthesis; U8: ubiquinol-8 biosynthesis; U9: ubiquinol-9 biosynthesis; U10: ubiquinol-10 biosynthesis; TA: total alkalinity; TDS: total dissolved solids. Mv-NsD1, Mv-NsD4, and Mv-NsD8: initial period; Mv-NsD96: mid period; Mv-NsD150: long period.

**Table 1 microorganisms-09-02315-t001:** Physicochemical characteristics of the mud samples.

Sampling Period		pH	ORP *	SC * (wt %)	TDS * (wt %)	TA * (mM)	Cl (ppm)	SO_4_^2−^ (ppm)	Total Sulfur (ppm)	Na (ppm)	Ca (ppm)	K (ppm)	Mg (ppm)	B (ppm)	Sr (ppm)	Ba (ppm)	Si (ppm)
Initial	Day 1	7.53	157	59.02	2.17	46.3	10,791	51.9	18.78	7829	109.26	60.07	40.95	44.23	14.56	3.78	6.10
	Day 4	7.66	124	56.81	2.01	-	11,415	63.9	19.49	8436	100.38	68.35	41.39	48.09	14.89	4.10	6.56
	Day 8	7.89	155	32.37	2.60	54.9	14,947	88.4	16.69	7538	38.82	60.59	33.93	44.58	11.13	2.93	3.15.
Mid	Day 22	7.71	88	41.39	3.11	56.5	16,114	53.6	19.79	11,256	66.33	82.75	51.69	63.07	17.12	4.69	4.88
	Day 96	8.42	95	1.05	3.57	60.8	18,343	13.9	30.71	13,301	72.67	113.96	62.71	83.92	20.06	6.37	7.57
Long	Day 158	8.01	−220	32.11	3.15	46.2	16,586	283.9	96.81	11,843	21.35	92.72	48.61	67.39	10.99	3.41	2.70

* ORP: oxidation/reduction potential; SC: solid content; TDS: total dissolved solids; TA: total alkalinity.

## Data Availability

The datasets generated during and/or analyzed during this study are available from the corresponding author upon reasonable request.

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
