# Peer review of "Assessment of Temporal Effects of a Mud Volcanic Eruption on the Bacterial Community and Their Predicted Metabolic Functions in the Mud Volcanic Sites of Niaosong, Southern Taiwan"

_microorganisms, 2021, doi:10.3390/microorganisms9112315_

Round 1
Reviewer 1 Report
In the manuscript “Assessment of temporal effects of the mud volcanic eruption over the bacterial community and their predicted metabolic functions in the mud volcanic sites of Niaosong, Southern Taiwan”, the authors describe the metabolic and community changes that inhabit volcanic areas as a function of geological changes the authors describe the metabolic and community changes that inhabit volcanic regions as a function of geological changes.
I suggest increasing the resolution of Figure 1I suggest increasing the resolution of Figure 1 e it would be useful to add the coordinates of the sampling site
The general quality of the figures must absolutely be increased, the figures are of low resolution
Author Response
Reviewer comments response
Dear editor,
We are grateful to the reviewer for his insightful comments on our paper. We have been able to include changes to reflect most of the suggestions provided by the reviewers.
Reviewer #1 Comments
In the manuscript “Assessment of temporal effects of the mud volcanic eruption over the bacterial community and their predicted metabolic functions in the mud volcanic sites of Niaosong, Southern Taiwan”, the authors describe the metabolic and community changes that inhabit volcanic areas as a function of geological changes the authors describe the metabolic and community changes that inhabit volcanic regions as a function of geological changes.
Comments and responses:
Comments:
Q1: I suggest increasing the resolution of Figure 1I suggest increasing the resolution of Figure 1 e it would be useful to add the coordinates of the sampling site.
Response: Thanks for the reviewer’s suggestion. We have increased the resolution of Figure 1 and added the coordinates as suggested.
Q2: The general quality of the figures must absolutely be increased, the figures are of low resolution.
Response: Thanks for the reviewer’s suggestion. The quality of all the figures in the manuscript has been improved as suggested.
Reviewer 2 Report
The manuscript is well-written, the experimental design and the temporal data presented are sound. The bioinformatics tools used with the statistical validation of the output are acceptable. I did not find any significant issues with the figures and the table except that the figure legends should be further elaborated to provide brief data outcomes and relevance to the study. The other comment is that the authors should consider elaborating overall how much methane and sulfur gas released by these and other volcanic muds to the atmosphere that would likely (or not) impact global climate change. This would improve the originality and novelness of the study. This would also improve the Conclusions section of this manuscript.
Author Response
Reviewer comments response
Dear editor,
We are obliged to the reviewer for his insightful comments on our paper. We have been able to include changes to reflect most of the suggestions provided by the reviewers.
Review #2 comments
The manuscript is well-written, the experimental design and the temporal data presented are sound.
Q1: The bioinformatics tools used with the statistical validation of the output are acceptable. I did not find any significant issues with the figures and the table except that the figure legends should be further elaborated to provide brief data outcomes and relevance to the study.
Response: Thanks for the reviewer’s suggestion. We have elaborated the figure legends wherever it is required.
Q2: The other comment is that the authors should consider elaborating overall how much methane and sulfur gas released by these and other volcanic muds to the atmosphere that would likely (or not) impact global climate change. This would improve the originality and novelness of the study. This would also improve the Conclusions section of this manuscript.
Response: Thanks for the reviewer’s suggestion. In this present study, we have only concentrated on the microbial abundance and the functional pathways of the mud volcanic site. Besides, we have analyzed the physicochemical characteristics of the mud fluids after various time intervals of the volcanic eruption. But, we have not planned to measure the methane and sulfur gas during emission in this particular study. Thus, we couldn’t elaborate or compare the gas emission status of this mud-volcanic site with others. We will consider this valuable suggestion during our future mud-volcanic site related studies.